# Biomaterials for Regenerative Medicine in Italy: Brief State of the Art of the Principal Research Centers

**DOI:** 10.3390/ijms23158245

**Published:** 2022-07-26

**Authors:** Francesca Camponogara, Federica Zanotti, Martina Trentini, Elena Tiengo, Ilaria Zanolla, Elham Pishavar, Elisa Soliani, Marco Scatto, Paolo Gargiulo, Ylenia Zambito, Stefano De Luca, Letizia Ferroni, Barbara Zavan

**Affiliations:** 1Translational Medicine Department, University of Ferrara, 44121 Ferrara, Italy; francescacamponogara@unife.it (F.C.); federicazanotti@unife.it (F.Z.); maritnatrentini@unife.it (M.T.); elenatiengo@unife.it (E.T.); elhampushavar@unife.it (E.P.); 2Medical Sciences Department, University of Ferrara, 44121 Ferrara, Italy; ilariazanolla@unife.it; 3Bioengineering Department, Imperial College London, London SW7 2BX, UK; solianie@icl.uk; 4Department of Molecular Sciences and Nanosystems, Ca’ Foscari University of Venice, Via Torino 155, 30172 Venezia, Italy; scatto@unive.it; 5Institute for Biomedical and Neural Engineering, Reykjavík University, 101 Reykjavík, Iceland; paologar@landspitali.is; 6Department of Science, Landspítali, 101 Reykjavík, Iceland; 7Chemical Department, University of Pisa, 56124 Pisa, Italy; zambitoy@unipi.it; 8Unit of Naples, Institute of Applied Sciences and Intelligent Systems, National Research Council, Via P. Castellino 111, 80131 Napoli, Italy; destefanol@unina.it; 9Maria Cecilia Hospital, GVM Care & Research, 48033 Cotignola, Italy; lferroni@gvmenet.it

**Keywords:** biomaterial, polymer, regeneration, tissue engineering

## Abstract

Regenerative medicine is the branch of medicine that effectively uses stem cell therapy and tissue engineering strategies to guide the healing or replacement of damaged tissues or organs. A crucial element is undoubtedly the biomaterial that guides biological events to restore tissue continuity. The polymers, natural or synthetic, find wide application thanks to their great adaptability. In fact, they can be used as principal components, coatings or vehicles to functionalize several biomaterials. There are many leading centers for the research and development of biomaterials in Italy. The aim of this review is to provide an overview of the current state of the art on polymer research for regenerative medicine purposes. The last five years of scientific production of the main Italian research centers has been screened to analyze the current advancement in tissue engineering in order to highlight inputs for the development of novel biomaterials and strategies.

## 1. Introduction

Regenerative medicine is the branch of medicine that effectively uses stem cell therapy and tissue engineering strategies to guide healing or replacement of damaged tissues or organs. In this scenario, biomaterials are crucial. In the last 20 years, the science of biomaterials has made advances in all fields from the development of full synthetic materials to green and nanostructured materials. Every novel biomaterial aims to have several advantages, including minimal antigenicity and fibrogenicity, the absence of local or systemic toxicity, infectious safety, hemocompatibility, a minimal infection-promoting effect, to serve as a substrate for cell adhesion and proliferation, to allow the deposition of the extracellular matrix (ECM), controlled degradability, mechanical resilience, ductility to processing, easy sterilization, and economically convenience.

The Italian research centers have proposed a wide variety of biomaterials ranging from polymers through ceramics to metals. The aim of this review is to provide a snapshot of the main Italian research centers that deal with the design, development, production and testing of biomaterials for regenerative medicine purposes. The last five-year scientific production of the main Italian research centers has been screened to provide an overview of the current state of the art on polymer applications.

## 2. Methodology

A PubMed search was executed for research articles on polymers for tissue engineering applications performed by Italian research centers. The search was done by searching the main Italian universities as “affiliation” together with the terms “polymer”, “hydrogel”, “alginate, “chitosan”, “hyaluronic acid”, “pectin”, “fibroin”, “graphene”, “poly(lactic acid)”, “poly(glycolic acid)”, “poly(lactide-co-glycolic acid)”, “poly(caprolactone)”, “poly(vinyl alcohol)”, “poly(urethane)”, or “decellularized extracellular matrix”. Only English-language original articles published between 2017 and 2022 were considered. All in vitro, in vivo, preclinical studies regarding the development, production, analysis, and application of polymers for tissue engineering and regenerative medicine purposes were considered. Systematic reviews and meta-analyses were excluded.

## 3. Polymers Overview

Polymers have significant importance in regenerative medicine due to their chemical and mechanical tunability. Polymers can be used for the production of cell culture systems, drug delivery systems, scaffolds, or biological substitutes (Table 1). A large amount of natural and synthetic polymers and composites is available for biomedical applications. Natural polymers include, for instance, alginate, collagen, gelatin, chitosan, fibroin, and hyaluronic acid. Major synthetic polymers are poly(lactic acid) (PLA), poly(glycolic acid) (PGA), poly(lactide-co-glycolic acid) (PLGA), poly(caprolactone) (PCL), poly(vinyl alcohol) (PVA) and poly(urethane) (PU).

Alginate is an anionic and hydrophilic polysaccharide that derives primarily from bacteria and brown seaweed. Genoa University (Department of Informatics, Bioengineering, Robotics and Systems Engineering) has developed microbeads made of alginate and chitosan with well-defined drug-loaded microvoids. These microbeads are a promising approach for the creation of biocompatible micro-structured scaffolds and for the development of a system with limited drug leakage [1].

At the Institute of Nanotechnology (NANOTEC, Lecce), various polymers and composites such as thiol-ene alginate [2], chitosan-based hydrogels [3,4], and PVA [5] have been tested as bioink for 3D printing. Semisynthetic hydrogels have been used for the 3D bioprinting of hepatic constructs to investigate drug-induced hepatotoxicity. This liver model can mimic the 3D microarchitecture of the hepatic tissue and be a valid alternative to animal models addressing some of the challenges and the limitations of non-physiological 2D culture systems [6]. In fact, 3D printing is a powerful technique that allows the precise and controlled deposition of biomaterials in a predesigned and reproducible manner. Furthermore, specific features of hydrogels allow for the creation of a thermo-sensitive system suitable for cell embedding and culturing and to obtain a bubble-free system with optimal chemical-physical characteristics for 3D culture systems [7].

The Institute of Electronics, Computer and Telecommunications (IEIIT) of Genoa proposed a 3D alginate-based hydrogel as an extracellular microenvironment to evaluate the effects of three-dimensionality on biological and immunological properties of neuroblastoma cells. This cancer model allowed the analysis of neuroblastoma growth, immunophenotype, and susceptibility on therapies improving the efficacy of personalized therapeutic approaches [8]. Instead, at the Italian Institute of Technology (IIT) of Genoa, sodium alginate/povidone-iodine film [9] and ε-caprolactone-p-coumaric acid copolymers [10] have been tested for the fabrication of dressings for wound and burn management. Both polymers successfully improved wound closure, demonstrating biocompatibility and healing properties. Moreover, at the Genoa University, membranes made of alginate or chitosan have been produced via electrospinning. This procedure relies on the application of a strong electric field to a polymer solution or polymer melt inducing the formation of nano- and/or micro-fibers [11]. The electrospun membranes could be used for several purposes, such as biomedical, pharmaceutical and environmental applications [12,13,14].

PVA is widely investigated by Padua University as a tailorable biomaterial for several tissue engineering applications. For instance, Conconi et al. indicated PVA as a promising biomaterial for the fabrication of artificial vessels [15]. PVA oxidation with potassium permanganate was investigated to realize scaffolds with controlled protein-loading ability, mechanical behavior, and biodegradability. Partial oxidation of PVA with potassium permanganate has proven to be an efficient method to fabricate smart scaffolds with biodegradation and protein delivery capacity [16]. Halogens such as bromine, chlorine, and iodine are less aggressive than potassium permanganate and perform controlled PVA oxidation, which prevents polymer molecular size degradation upon chemical modification. Halogen oxidized PVA hydrogels exhibited high biocompatibility in vitro as well as in vivo, resulting in neither cytotoxic nor inducing severe host immune reactions compared to non-toxic PVA (Figure 1) [17]. Oxidized PVA was cross-linked with a decellularized intestinal wall to develop a composite scaffold for intestinal restoration. In vitro and in vivo analyses proved cell adhesion and proliferation and the development of organized and continuous tissue walls around the support with a connective appearance [18]. Moreover, a hydrogel-hybrid scaffold made of PVA and decellularized Wharton’s jelly ECM was studied as a scaffold for cartilage restoration [19,20].

Chitosan, mentioned above, is a biopolymer derived from the partial or total deacetylation of chitin, the second most abundant polysaccharide. At Trieste University, many studies have been conducted to characterize the properties of chitosan. This biopolymer possesses peculiar biological and rheological features, and in the presence of diol-binding and cross-linking molecules it is possible to generate mechano-responsive biomaterials for tissue engineering [21,22,23]. Instead, the Food and Drug Department of Parma University has proposed 3D-printed chitosan as a scaffold for human fibroblast growth [24], as a tool for epithelia regeneration [25], and for wastewater treatment [26]. In orthopedics, metals are the most common materials used to produce different types of medical devices, and several polymers are used to coat them to improve biocompatibility [27]. The Department of Industrial and Digital Innovation of Palermo University has developed a bio-coating via the galvanic deposition of chitosan, hydroxyapatite and collagen on stainless steel. The results have demonstrated that the composite bio-coating slows down the corrosion rate of steel and improves the biocompatibility of the metal device [28,29].

Trento University (Department of Industrial Engineering and Biotech Research Center) has conducted several studies on silk fibroins. The silk fibroin has been tailored modifying the secondary structure and molecular weight by linking specific active groups to the amino acid side chains [30]. Silk fibroin is a perfect candidate for the production of biomedical prostheses and tissue engineering scaffolds because it can be fabricated as films, sponges, fibers, nets or gels with predictable degradation times [31,32]. The silk of a spider was also investigated, and the intrinsic mechanisms of adhesion and the role of water and humidity in the spider’s silk adhesion were analyzed. Results demonstrated that cribellate threads operate best when both the nanofibrils and the axial fibers contribute to adhesion. In the future, the design of materials for tissue engineering and regenerative medicine could consider the model of cribellate thread adhesion [33,34].

Hyaluronic acid is the chief component of the connective tissue that forms the gelatinous matrix around cells. The Institute of Polymers, Composites and Biomaterials of National Research Council (IPCB-CNR) of Naples is specialized in hyaluronic acid composites and derivates. For instance, in collaboration with Rome University, hyaluronics acid hydrogels have been proposed for the treatment of degenerative meniscus lesions or intervertebral disc degeneration [35,36]. In the first case, they have evaluated the clinical efficacy and beneficial effects of the treatment with hyaluronic acid hydrogel injected in patients for the conservative treatment of the meniscal lesion. Preliminary findings suggest that the use of hyaluronic acid in the conservative management of degenerative meniscus lesions is clinically effective and enhances meniscus healing and reduces arthroscopic partial meniscectomy at one-year follow-up [35]. In the second case, they have tested hydrogels composed of hyaluronic acid and platelet-rich plasma (PRP) as a carrier for human mesenchymal stem cells (MSCs) for intervertebral disc regeneration. By a disc organ model, it has been demonstrated that the composite hydrogel promotes MSCs engraftment and differentiation and integrates with the surrounding tissues. However, other studies are needed to provide further evidence for intervertebral disc regeneration [36].

Hydrogels derived from the decellularized extracellular matrix (dECM) have been investigated at the Veneto Institute of Molecular Medicine (VIMM, Padua). Different dECM hydrogels with tissue-specific signatures have been tested to support organoid culture and muscle regeneration [37,38,39]. The myogenic potential of dECM obtained from bovine pericardium have been evaluated in vitro using murine muscle cells. Results indicated that ECM proteins are a promising tool for skeletal muscle regeneration [40]. In addition, membrane of platelet-rich fibrin was investigated at Padua University. Leukocyte-fibrin-platelet membranes have been prepared through multiple cycles of apheresis. The membrane showed high elasticity and deformation capacity due to the high fibrinogen content. The high content of leukocytes, monocytes/macrophages, and platelets sustained the local release of bioactive molecules including platelet derived growth factor, vascular endothelial growth factor, interleukin-10, and tumor necrosis factor alpha [41,42].

The Department of Biotechnology and Biosciences (Milan University) and the Milan-Bicocca University have functionalized the gelatin polymer to improve mechanical stability at physiological temperatures. Cross-linked gelatin has been generated by homobifunctional triazolinediones [43], and homobi- and homotri-functional tetrazoles in order to obtain more stable hydrogels [44]. Instead, the Polytechnic of Milan studied pectin hydrogel for tissue engineering applications. Cross-linked pectin can form hydrogels that can retain a large amount of water [45] that can be used in 3D culture systems for studying cell behavior [46].

The Basilicata University is focused on recombinant protein-based biomaterials for biomedical applications [47]. Self-assembling polymers have been produced by combining block polypeptides with genes encoding polypeptide sequences. Thanks to their properties, self-assembling polymers are useful in the integration of synthetic devices in the body preventing the invasion of cells [48,49]. Short peptide hydrogels are attractive biomaterials but typically suffer from limited mechanical properties. To overcome this drawback, nanomaterials can be embedded to reinforce and confer additional physicochemical properties. The family of carbon nanostructures comprises several members, such as fullerenes, nano-onions, -dots, -diamonds, -horns, -tubes, and graphene-based materials [50,51]. The interaction between the two-amphiphilic components, unprotected tripeptide and oxidized nano-carbons, provide supramolecular hydrogels at physiological conditions that are useful for the design of composite biomaterials with advanced properties [52].ijms-23-08245-t001_Table 1Table 1Characteristics and applications of polymers.MaterialsCharacteristicsApplicationsRef.AlginateMicro-structured hydrogel Bioprintable Accelerate wound closureDrug delivery Cancer model (neuroblastoma) Wound healing[1,2,9,12]ChitosanChitin derivative BioprintableSoft tissue engineering Mechano-responsive biomaterial Wound healing[3,4,13,14,21,22,23]dECM hydrogelsContains growth and differentiative factors Myogenic potentialOrganoid culture Skeletal muscle tissue engineering[37,38,39,40]GelatinCollagen derivative Mechanically stableDrug delivery Cell culture Tissue engineering[43,44]Hyaluronic acidComponent of connective tissue Injectable Cell culture Treatment of meniscal lesions Intervertebral disc regeneration Wound healing[35,36]PectinPlant derivative Easily available SuperabsorbentCell culture Tissue engineering[45]PVATailorable by oxidationDrug delivery Composite scaffold[16,17,18,19,20]Self-assembling peptidesVersatileCombined with other biomaterials for improving bioactivity[47,48]Silk fibroinTailorable through modification of molecular weight and functionalization.Small vessels and nerve scaffolds[30,31,32]


## 4. Application Fields

### 4.1. Neurological Applications

In biomedical neuroscience, biomaterials are used for regeneration, healing, the development of the central nervous system (CNS), and nerve grafting in peripheral nerve injuries. Polymers used for these purposes vary from synthetic biodegradable biomaterials, such as PLA and PCL, carbon-based materials such as carbon nanotubes (CNTs) and graphene oxides (GO), and natural biomaterials such as fibroin and chitosan (Table 2).

Synthetic materials exhibit a higher degree of purity compared to natural-based products; however, they are often less biocompatible [53]. The Neurobiology Sector of the International School for Advanced Studies (SISSA/ISAS) of Trieste is a leader in developing neurologically compatible synthetic nanomaterials for medical purposes. This institution has focused its work on GO derivatives and their impact on the nervous system for neurodegenerative disorders and synapse-based therapeutic interventions (Figure 2) [54]. GO is a hydrophilic carbon material, structured as a 2D network with interesting physiochemical features such as its strength, its light weight, its high conductivity and its chemical stability. It is suitable for biomedical purposes, as it can be functionalized for improved molecular interactions [55]. GO nanosheets (nanoscale-sized) were used for the manipulation of excitatory neuronal transmission. An excessive glutamate response in the CNS is linked to neuropsychiatric disorders: GO decreased glutamatergic activity by targeting the release of presynaptic vesicles in the hippocampus [56,57]. In addition, through both in vitro and in vivo models, the role of GO-nanosheets/glutamate interference in the amygdala synaptic circuitry has been explored. An excessive glutamatergic activity in the lateral nucleus of the amygdala complex is involved in anxiety and post-traumatic stress disorder [58,59].

CNTs were investigated on nerve conduits and axon regeneration [60,61,62]. Single and multi-layer CNTs, as GO, exhibit good conductivity and biocompatibility [63]. At the neural cell interface, they can potentiate synaptic networks’ formation and function [64]. Thus, artificial multiwall 3D carbon nanofibers (CNF) were developed to mimic the spinal environment’s mechanical and electrical properties by promoting synapse formation, axonal growth, and excitability [60]. CNF allowed fiber regeneration in the lesion site and sped motor function recovery in rats with spinal cord injuries [61]. While transparency is not a pivotal characteristic for nerve conduits, it is nonetheless useful for the correct positioning of the graft during surgery, and for monitoring pre-clinical assays [53]. Research led by Pampaloni et al. explored a new methodology to synthesize transparent CNTs by growing them on fused silica surfaces, therefore creating optimal biomaterials for tissue engineering [62]. This work on CNT production methods also comprises the development of a chemical vapor deposition process, a cost-effective technique that increases the possibility of CNT applications [64]. Lastly, polystyrene/carbon-nanotube (PL-CNT) nanocomposites have been developed for the creation of implantable devices with suitable neural interfaces. PL-CNTs stimulated cultured primary neurons electrically and could therefore be incorporated in neuro-prosthetic and neurostimulation devices [64].

PCL is a very resistant material that is primarily used for the replacement of hard tissues. However, its molecular weight can be adjusted for soft tissue engineering and nerve repair [65]. At the Organic Synthesis and Photoactivity Institute (CNR-ISOF) of Bologna, Saracino et al. demonstrated the compatibility of electrospun fibers of PCL with astrocytes in vitro [66]. When in the proximity of PCL fibers, astrocytes did not exhibit toxicity, although they re-arranged their structure [66].

Oxidized PVA has been studied by for injured peripheral nerve regeneration at Padua University. Hollow nerve conduits made of PVA partially oxidized with potassium permanganate or bromine were investigated as delivery systems of ciliary neurotrophic factor. In vitro and in vivo experiments demonstrated that the functionalized oxidized PVA conduits represent promising strategies to enhance peripheral nerve regeneration by guiding axon growth while delivering therapeutic neurotrophic factors [67,68,69]. In addition, oxidized PVA has been modified to produce a biodegradable wrap for peripheral nerve injuries. Tests on rat sciatic nerve transection and neurorrhaphy showed favorable lesion recovery after 12 weeks of treatment with the oxidized PVA-based nerve wrap [70,71].

Several Italian groups have directed their research towards the production of biomaterials with natural polymers. The purification process of natural hydrogels and fibers might be more complex; however, raw materials are abundant and have better biocompatibility on average. The Department of Clinical and Biological Sciences of Turin University worked on mechanisms of nerve regeneration and Schwann cells differentiation using a variety of scaffolds. The team has optimized the hydrogel preparation process and composed blends of natural polymers for optimal drug incorporation. The obtained material has been tested for the controlled release of VEGF, crucial in tissue repair and angiogenesis. Beneficial effects were observed on motor neuron survival, Schwann cell proliferation and the sustained growth of regenerating peripheral nerve fibers [72]. Some of their recent studies for peripheral nerve regeneration have focused on different mixes of injectable gelatins (obtained from collagen thermal degradation) to support Schwann cell migration and growth [73], natural hydrogels as fillers in peripheral nerve conduit channels [74], and decellularized nerve allografts [75].

Chitosan is another popular natural material for neural grafts. When peripheral nerves are repaired with chitosan grafts, their axonal growth rate increases and scarification is drastically reduced [76]. Chitosan hollow tubes enriched with skeletal muscle fibers provide an alternative to the muscle-in-vein conduits that are commonly used. The result is an easy to manage graft, which also induced a favorable environment for Schwann cell migration, thanks to the release of Neuregulin 1 factor by the muscle tissue [77]. In addition to collagens and chitosan, another natural biomaterial commonly used in nerve regeneration is silk. Silk fibroin is a complex of fibrous proteins synthesized by the glands of arthropods, such as silkworms and spiders. Their biocompatibility is well-established, and they exhibit extremely valuable mechanical properties for applications as neural conduits: they are elastic, flexible and highly resistant to stretching/compression. Among the several Italian research groups working on silk fibroins, Silk Biomaterials Srl has patented SilkBringe^TM^, a two-layered silk (*Bombyx mori*) nerve conduit obtained through the electrospinning technique. The novelty of SilkBringe^TM^ lies in its architecture, as it is functional yet structurally solid; while silk itself is responsible for biocompatibility and strength [78,79]. The Department of Clinical and Biological Sciences (Turin University) and the Neuroscience Institute of the Cavalieri Ottolenghi Foundation (NICO) in pre- and post-clinical trials have investigated the product’s efficacy. These studies validated SilkBringe^TM^, safety and its beneficial effect on median nerve recovery [80,81]. These groups are specialized in nerve tissue engineering and regeneration: they have produced several works on decellularized nerve grafts [82], neuronal grafts conduits based on synthetic biomaterials (PLC and polyhydroxyalkanoates) [83,84], and other nerve regeneration techniques [85,86].

Lastly, the CNR Institute on Membrane Technology (CNR-ITM, Cosenza) has worked on 3D culture for liver regeneration and neuroregenerative biomaterials [87,88,89]. This team investigated artificial 3D membranes for the development of biomimetic models, reliable tools for in vitro observations of neurological diseases [90]. In the last decade, several membrane bioreactors for neural differentiation have been developed including, PCL and chitosan for neural spheroids growth, PLLA microtube arrays and PLGA membrane platforms [91,92,93,94,95]. PLA is a hood term that comprehends three different biodegradable, miscible blends of enantiomers: poly L-lactictide (PLLA), poly D-lactictide (PDLA) or the stereocomplex poly lactictide (SC-PLA) [96]. The composition of each blend determines the mechanical properties of the final material. PLGA is also derived from PLA, as it is a co-polymer of PLA and polyglycolide (PGA) that can be synthesized with varying lactide/glycolide ratios [97]. Interestingly, the PLGA matrix array created by the CNR-ITM group was suitable for neural growth and induced the formation of proper cell adhesion and cellular contact formation among cells [94,95]. Moreover, the PLLA microtube array membranes offer a more controllable growth for neural development and orientation, and could be used in preclinical research as an important investigative tool for neurodegenerative pathologies [93]. Neural differentiation also occurred on poly-butylene 1,4-cyclohexane dicarboxylate (PBCA) film. The Centre of Excellence on Innovative Nanostructured Materials (CEMIN) of Perugia University used PBCA as a substrate for MSC neural differentiation [98] and has worked with functionalized PLLA biomaterials suitable for MSC interactions [99].ijms-23-08245-t002_Table 2Table 2Characteristics and applications of biomaterials in the neurological field.MaterialsCharacteristicsApplicationsRef.ChitosanInduce axonal growth Reduces tissue scarificationNerve grafts conduits[76,77]CNTMechanical strength Thermal inertnessTransparent nerve conduits Axon regeneration Spinal cord regeneration Implantable devices with suitable neural interfaces[62,63,64]FibroinElastic Flexible Highly resistant to stretching/compressionNerve grafts conduits Recovery of the median nerve[78,79,80,81]OGMechanical strength Electrical and thermal conductivityEducation of neural connections Guide neural growth and differentiation[52,53,54,55,56,57,58,59]PBCABiocompatibleNeural differentiation Cell culture[98]PCLTailorable for soft tissue engineering Biocompatible for neural tissue (astrocytes)Nerve grafts conduits[65,66]PLABioabsorbableTissue growth Cell culture Neural differentiation[92,93]PLGABiodegradableTissue growth Cell culture[95]PVATailorable by oxidationHollow nerve conduits Nerve wrap[67,68,69,70,71]

### 4.2. Cardiovascular Applications

Cardiac-associated diseases are one of the most common causes of mortality globally. Current treatments for these diseases (e.g., cell therapy, heart transplantations, bovine or porcine heart valve implantations and prostheses) unfortunately suffer from several limitations, such as lack of donors, immune rejection, coagulant therapy, and limited durability. Therefore, research in the cardiovascular field leans on tissue engineering for the development of therapies that will replace the current standards [100].

The principal centers involved in cardiovascular tissue engineering are mainly distributed in Veneto, Tuscany, Lazio, Abruzzo and Piedmont. In recent years, dECM, which is the noncellular component of tissue that retains relevant biological cues for cells, has been widely investigated in tissue engineering applications because of its essential role in guiding muscle regeneration (Table 3). LIFELAB (Living, Innovative, Fully Engineered, Long-lasting and Advanced Bioreplacement), a program of advanced bioreplacement that collaborates with Padua University, is focused on decellularized organs, a very plastic matrix that could be adopted for in vivo tissue engineering and the production of tissues and organs for transplant [101,102,103,104,105,106]. Using dECM, novel solutions have been created for the development of biomaterials based on biological tissue which could be suitable for applications in cardiovascular repair, corrective, and reconstructive surgery [107,108,109]. The Department of Medicine and Aging Sciences (Chieti-Pescara University) and the Department of Health Sciences, (University of Eastern Piedmont Amedeo Avogadro, UniUPO) also work on dECM to implement a promising regenerative strategy for cardiovascular diseases. They have found that the use of extracellular matrix proteins, as biomaterial supports, could represent a promising therapeutic strategy for cardiac muscle tissue engineering (Figure 3) [110,111,112].

Biocompatibility Innovation (BCI) is a company located in Este (Padua, Italy) specializing in biological implantable medical devices from animal origins. BCI has tested different biomaterials to improve bioprosthetic heart valve (BHVs) tissues and enhance their stability whilst ensuring a satisfactory degree of immunological tolerance [113,114]. The University of Verona (Department of Surgery, Dentistry, Pediatrics & Gynecology) is instead specializing in evaluating the efficacy of silk fibroin for cardiovascular tissue engineering and regeneration purposes in vitro. Silk allows the development of tubular scaffold/small caliber vascular grafts, and is commonly used for the substitution, repair, and regeneration of blood vessels [115,116].

Moreover, Tuscany hosts three different groups that deal with cardiac tissue engineering applications, and they are all located in Pisa. First, the Institute for Chemical and Physical Processes has produced a bioartificial 3D cardiac patch with cardioinductive properties on stem cells. The studies focus on biocompatible PLGA and gelatin-based polymeric material for cardiovascular surgery [117,118]. Secondly, the Research Center “E. Piaggio” and 3R Center (Department of Civil and Industrial Engineering) of Pisa University work on ameliorating biomimetic patches for myocardial repair, which is used for providing a 3D structural support for cellular growth during new tissue formation (Figure 4) [119,120]. In addition, they developed novel peptide-modified scaffolds as biomimetic matrices; these scaffolds mimic the biomolecular signals of the ECM, improving cardiac progenitor cell adhesion and differentiation toward myocardial phenotypes [121].

The National Italian Agency for New Technology, Energy and Sustainable Economy, Diagnostics and Metrology Laboratory (ENEA, Rome), and the Interdepartmental Research Centre for Regenerative Medicine (CIMER, University of Rome) have found different approaches using tissue engineering to solve severe muscle injures and other cardiac damage, such as post-ischemic myocardial insult. They have proposed the use of a scaffold serving as an artificial ECM in cardiac tissue engineering. The ECM hosts the cells and improves their survival, proliferation, and differentiation, enabling the formation of new tissue. Therefore, thanks to its properties of stimulation neomyogenesis and vascularization, it should be considered as a valuable biomaterial to be used to fabricate the tissue-specific three-dimensional structure of interest to promote muscle regeneration and repair [122,123].

Lastly, the Department of Translational Medicine of Ferrara University has functionalized an elastomeric scaffold with exosomes derived from MSCs for the contractile restoration of myocardial scars. Exosomes are natural nanoparticles that show several advantages compared to other engineered synthetic nanoparticles thanks to their cell-based biological structures and functions. A combination of elastomeric membrane and exosomes was obtained and tested as a bioimplant, and their results confirm that exosomes, entrapped onto elastomeric scaffolds, increase wound healing properties [124].ijms-23-08245-t003_Table 3Table 3Characteristics and applications of biomaterials in the cardiovascular field.MaterialsCharacteristicsApplicationsRef.dECM3D structure Heterogeneity Mechanical support Suitable microenvironmentRegeneration of skeletal muscle tissue Stimulation of myogenesis and angiogenesis Scaffold Restoration of damaged organs[103,104,105,106,110,111,112]GelatinStructural support Cardio-inductivity Slow biodegradation rate Rheological propertiesBiomimetic cardiac patch Drug delivery Tissue engineering of skeletal muscle[118,120]PLGAAdaptable structureDrug delivery systems Bioartificial 3D cardiac patch Suitable for cardiovascular cell growth[117,118]Silk fibroinBiocompatibility Textile layers Cellular adhesiveness Low immunogenicityDevelopment of vascular grafts/tubular scaffolds Promotion of vascularization Repair of skin wound[115,116]

### 4.3. Skeletal Muscle Reconstruction

Acellular matrices are biocompatible and non-immunogenic materials commonly used in muscle regeneration. ECM is very heterogeneous, made of several components secreted by resident cells, functional and structural proteins, glycosaminoglycans, glycoproteins, and other small molecules [125]. Padua University investigated the use of decellularized human skeletal muscle for regenerative purposes, including abdominal wall-defect restoration [126,127,128], vessel replacement [129,130], and tracheal and esophageal reconstruction [131,132]. Recently, the diaphragmatic muscle has been decellularized using a detergent-enzymatic treatment in order to create a scaffold with functional features. In particular, pediatric human muscle precursors have been seeded inside the ECM scaffold, demonstrating that the construct can activate regenerative response in vitro promoting cell self-renewal and positive ECM remodeling [133]. Moreover, a human decellularized diaphragm has been tested as scaffold material for volumetric muscle loss treatment in mice [134]. The detergent-enzymatic treatment did not affect the expression of Collagen I and IV and Laminin, while causing the loss of HLA-DR expression to produce non-immunogenic grafts. The acellular diaphragmatic grafts did not elicit a severe immune reaction, integrating with the host tissue (Figure 5) [134]. Meanwhile, Boso et al. developed a porcine diaphragmatic dECM-derived hydrogel for diaphragmatic applications. They obtained a tissue-specific biomaterial able to mimic the complex structure of skeletal muscle ECM that can be used as a stand-alone patch that is useful for diaphragmatic muscle defect repair [135].

As mentioned before, one of the most important new approaches in biomaterial applications has been identified in 3D printers and their potential use to create multiple types of scaffolds. The BioRobotics Institute of Sant’Anna School University (Pisa) made a 3D PU-based soft porous scaffold functionalized with ECM components such as laminin and fibronectin. The results showed an increase and improvement in myoblast proliferation [136]. ENEA in collaboration with the Interdepartmental Research Center of Regenerative Medicine at Tor Vergata (Rome University) worked on a 3D printing decellularized matrix for volumetric muscle loss, obtaining promising results in muscle regeneration [137,138]. In addition to that, as previously mentioned, they studied biofunctional composite electrospun biomaterials that mimic the 3D microstructural arrangements of muscle tissue. They proposed the surface biofunctionalization of the electrospun scaffolds via click chemistry to modify the surface of the fibers with selected growth factors or peptides to guide skeletal muscle regeneration [139].

### 4.4. Skin Regeneration

In the last years, many Italian teams have focused on skin regeneration and have developed several types of skin substitutes from artificial and natural materials. Engineered skin substitutes can be developed from acellular materials or can be synthesized from autologous, allograft, xenogenic, or synthetic sources. Several types of cells including native immune response cells, endothelial progenitors, keratinocytes, and fibroblasts are required and activated to allow skin wound healing. Based on these considerations, many strategies have been developed for skin regeneration [140]. These methodologies can be divided in conventional or novel approaches for tissue engineering (Table 4).

Genoa University has studied a new type of medication that combines alginate, silk sericin and platelet lysate in a freeze-dried sponge and is able to generate a bioactive wound dressing to care for skin lesions. This biomembrane was tested on a murine model of skin lesions, demonstrating that the combination of these elements can promote the healing process, inducing an accelerated inflammation as the initial step for regeneration, the formation of granulation tissue and the deposition of new collagen (Figure 6) [141]. In a previous work, the same group discovered the potential use of platelets to improve wound healing, testing a bioactive membrane functionalized with PRP. They used this membrane on a diabetic mice chronic ulcer model, demonstrating that this device was able to induce wound healing by increasing the thickness of the regenerated epidermis and vessel number [142]. As mentioned above, the conventional surgical treatments with autograft or allograft skin have been used to provide a life-saving skin replacement on the damaged area. With the aim of improving the outcome of the patient and enhancing the skin wound healing, the use of biological acellular dermal matrices (ADMs) was recently introduced. ADMs are biological scaffolds that maintain the structural and biochemical properties of the ECM after the removal of the cellular components. The potential of this device was demonstrated by several research groups such as the Burns Centre and the Emilia Romagna Regional Skin Bank, which worked on the use of the human dermal matrix (HDM) in combination with grafted skin for the treatment of full-thickness skin wounds [143]. Cell-free scaffolds derived from the human dermis can induce and speed up skin regeneration, as well as after mastectomy in breast cancer patients [144].

PLGA, thanks to its low toxicity and biocompatibility with tissue and cells, is used for devices that come in contact with the skin [145]. The Centre for Advanced Biomaterials for Health Care of IIT and the Department of Chemical, Materials and Industrial Production Engineering (University of Naples Federico II) have designed microneedles based on PLLA/PLGA microparticles and a PVP-made tip for controlled intradermal drug delivery. These devices are biodegradable and compatible with skin tissue both in vitro and in vivo [146,147,148]. Their work is also focused on the development of drug-delivery systems based on PLGA, PCL, PEG and other hydrogels [149,150,151,152,153] (Table 5).

The Institute of Applied Sciences and Intelligent Systems (CNR-ISASI) in Naples has recently developed hybrid flexible devices for sensing and transdermal drug delivery based on PEG diacrylate (PEGDA) hydrogels having different molecular weights (from 250 to 10,000 kDa) [154,155]. By combining the polymeric matrix with inorganic materials with suitable optical properties, such as gold or titania nanoparticles, the ISASI researchers fabricated sensors for the detection of specific analytes with label free optical techniques [156,157,158,159]. The hydrogels, added with a proper photoinitiator, could also be lithographically casted in microneedle arrays this ensure the controlled release of active molecules through the skin [160,161,162].

### 4.5. Bone Regeneration

The standard practices in bone defect repair involve autografts or allografts from a donor or cadaver. Both types of grafts promote the release of several bioactive molecules such as bone morphogenetic proteins and other growth factors that facilitate the differentiation of osteoprogenitor cells and MSCs [163]. Both techniques have the purpose of inducing osteogenesis, but there are some disadvantages, including immunogenicity, the quality and supply of bone grafts, and compatibility [164]. On the contrary, bone tissue engineering (BTE) aims to regenerate bone defects without bone grafts and therefore without additional complications, such as donor site morbidity, immunogenicity, and poor vascularization. To create a good environment to promote osteogenesis, BTE involves several components: (a) osteogenic cells to generate a novel bone tissue matrix; (b) biocompatible scaffolds made of bioactive materials that mimic bone ECM (e.g., polymeric and/or ceramic scaffold); (c) bioactive molecules leading morphogenetic signals; and (d) vascularization for nutrients and waste transport. Therefore, a perfect biomaterial for BTE should induce osteogenesis promoting cell differentiation into osteoblasts, support the 3D growth of new bone tissue, and integrate with surrounding bone tissue [163]. These goals need to be taken into consideration during novel biomaterial development. Polymers are largely employed in BTE because they can be functionalized with organic and bioactive molecules or can be used to coat metal implants improving osteointegration (Table 6).

The Department of Industrial Engineering of Padua University has investigated the biofunctionalization for chitosan and wollastonite and diopside to improve bone integration [165,166]. A chitosan-based scaffold has been functionalized with a different sequence of cell adhesive peptides via covalent conjugation to increase human osteoblast adhesion and proliferation [165]. Instead, a wollastonite and diopside-based scaffold has been functionalized with adhesive peptides mapped on vitronectin to improve external and internal cell colonization, the formation of new blood vessels, and the stimulation of ectopic mineralization [166]. Conversely, Porrelli et al. (Trieste University) have tested chitosan as a bioactive coating. Porous alginate/hydroxyapatite scaffolds were functionalized with lactose-modified chitosan. The scaffolds were stable, homogeneously coated with chitosan, and supported the adhesion, proliferation and differentiation of MSCs [167]. Di Liddo et al. (Padua University) functionalized a matrix made of PCL, hydroxyapatite, and bone ECM with alginate threads to produce a bone substitute with interconnected pores and a trabecular bone-like structure [168].

In Venice, the company Nadir—Plasma & Polymers produces biocompatible polymers filled with antibiotics to prevent infection during bone regeneration. The functionalized polymer can be 3D printed to produce antimicrobial scaffolds for MSCs differentiation into osteoblasts, allowing matrix mineralization and osteogenic marker expression [169]. Different research groups have explored 3D printing technologies to create scaffolds that reproduce the physiological environment of bone tissue in order to facilitate regeneration events. For instance, the Polytechnic of Turin worked on collagen-based inks for 3D printing bone scaffolds. Nano-sized mesoporous bioactive glasses enriched with strontium ions have been combined with type I collagen. The 3D-printed scaffold was able to induce the osteogenic differentiation of stem cells and to accelerate the mineralization kinetic in animal models [170,171]. A bioactive glass-ceramic was surface-functionalized with a blend of type I collagen and PU. In vitro human tests with periosteal derived precursor cells have demonstrated that the polymer-coated material was a good substrate for cell adhesion and growth, an effective solution to mimic the composite nature of bone ECM [172].

The Institute of Science and Technology for Ceramics of the National Research Council (ISTEC-CNR, Faenza) have enveloped strategies and calcium phosphate-based scaffolds for bone tissue reconstruction. A particular statistical tool has been used to design the biofabrication of hybrid hydroxyapatite/collagen scaffolds for bone regeneration and optimize their integration in a multilayer osteochondral device. The statistical tool allows for the selection of hydrogel concentration, hydroxyapatite/collagen ratio and cross-linker content. With these input parameters, multi-layer scaffolds have been synthesized with a graded mineralization rate that can be used to mimic the whole cartilage-bone interface [173]. Instead, biomorphic calcium phosphate bone scaffolds (GreenBone™) featuring osteon-mimicking, hierarchically organized, 3D porous structures and lamellar nano-architecture have been proposed for the regeneration of load-bearing segmental bone defects. In comparison with allografts, these novels scaffold showed higher new bone formation and quality of regenerated bone in term of thickness and number of trabeculae [174]. Moreover, at Padua Unversity, the osteoinductivy of calcium phosphate has been combined with the intrinsic immune-characteristics of GO in a biocompatible nanomaterial called maGO-CaP (monocytes activator GO complexed with CaP) [175]. Itis known that activated monocytes can communicate pro-osteogenic signals to MSCs and promote osteogenesis. maGO-CaP have enhanced the expression of key osteogenic pathways in MSCs, and when injected in murine tibia of mice have boosted local bone mass and the bone formation rate [175].

Calabrese et al. (Messina University) have tested innovative nanostructured scaffolds for BTE with MSC cultures. Mg/hydroxyapatite/collagen type I scaffolds have been nanofunctionalized with gold nanorods, palladium nanoparticles, or maghemite nanoparticles. Calabrese’s results are in agreement with previous studies that found cytotoxic effects for both gold nanorods and palladium nanoparticles. Instead, the high intrinsic magnetic field of superparamagnetic nanoparticles had improved both osteoconductivity and osteoinductivity in MSCs [176]. Likewise, the Istituto Ortopedico Rizzoli (IOR, Bologna) investigated the superparamagnetic nanoparticles to improve ostintegration. PCL-based scaffolds added with hydroxyapatite and different concentrations of superparamagnetic iron oxide nanoparticles (SPION) were fabricated by 3D printing [177]. Microscopy analysis confirmed a homogenous distribution of hydroxyapatite and SPION throughout the surface. MSCs seeded onto PCL-based scaffolds have shown good proliferation and intrinsic osteogenic potential. Instead, Pavia University has investigated the effects of pulsed electromagnetic fields (PEMF) on human osteoblast-like cells seeded on wool keratin scaffolds. Wool keratin scaffolds were very stable with a low degradation rate, and PEMF improved cell proliferation and differentiation, and the production of calcified bone ECM. This system could support long-term bone regeneration in vivo [178].

At Bologna University, synthetic polymer-based scaffolds doped with calcium silicates (CaSi) have been investigated. Highly porous scaffolds have been prepared combining PCL or PLA, dicalcium phosphate dihydrate (DCPD) and CaSi. The scaffolds had internal open porosity, which can be colonized by cells, release biologically active ions, and create nucleated apatite [179]. In vitro tests with MSCs have demonstrated new vessels formation into both PCL- and PLA-based scaffolds. The vascularization of bone substitute is a crucial factor for his stability, therefore the formation of new vessels is desirable for a potential use in bone regeneration [180]. Mineral-doped scaffolds, composed by PLA, DCPD and CaSi, have also been tested as a substrate for growth and the proliferation of human periapical cyst MSCs [181]. These cells have recently been discovered and collected from inflammatory periapical cysts, and they are a biological waste in dentistry. The mineral-doped scaffolds colonized with these autologous stem cells could represent a promising strategy for the regenerative healing of periapical and alveolar bone [181]. Lastly, in collaboration with Ferrara University, these porous mineral-doped PLA-based scaffolds have been enriched with exosome [182]. Exosomes are involved in several physiological processes including neo-angiogenesis and bone homeostasis. Results demonstrated that exosomes firmly entrapped on scaffold surfaces improved osteogenesis stimulating type I collagen, osteopontin, osteocalcin, and osteonectin in MSCs [182].

Titanium and its alloys are commonly used for bone regeneration thanks to their biocompatibility and favorable mechanical properties. Nevertheless, porosity and coating are generally managed to improve the osteointegration of titanium-based implants. Randomized trabecular titanium structures obtained by the 3D printing technique have been investigated at Milan University [183]. The histomorphometric and biomechanical tests showed a fast osseointegration of the trabecular titanium structure both in cortical and in cancellous bone of a sheep model. The bone ingrowth has included increased non-mineralized matrix deposition and enhanced mineralization that is complete after 14 weeks [183]. The IOR has also fabricated bone implants using 3D printing and made of titanium or cobalt-chrome-molybdenum alloy [184,185]. Two types of coatings (hydroxyapatite or type I collagen) have been tested to improve the osteointegration into lateral femoral condyles of rabbits. Hydroxyapatite coatings have hastened the bone-to-implant contact process and mineral apposition rate, while collagen did not significantly improve the osteointegration process [184]. Moreover, a coating made of a partially sulphated hyaluronic acid functionalized with a dopamine moiety (sHA-DA) has been tested to prevent acute bacterial growth in an in vivo model of highly contaminated titanium implant. Titanium nails coated with sHA-DA have been placed in the femoral medullary cavity of rabbit for 12 weeks. After one week, only the animals treated with sHA-DA-coated nails did not show systemic or local bacterial infection. In addition, nails were histocompatible and allowed bone growth in adhesion to the metal surface [185]. Calabrese et al. coated titanium with titanium dioxide and ferric oxide to inhibit bacterial growth and enhance osteointegration. Nano-functionalized titanium substrates exhibited a good antibacterial activity towards *Staphylococcus aureus* and MSC proliferation and differentiation [186]. Padua University has instead tested sphene biocoating to improve titanium osteointegration [187,188]. In vitro tests have demonstrated the adhesion, proliferation and osteogenic differentiation of human MSCs [187]. Moreover, peri-implant bone healing was histologically and histomorphometrically evaluated into proximal femurs of rat models. Delamination of the coating has occurred in some cases, and no synovial fluid has been collected on the test side confirming sphene biocompatibility [188].ijms-23-08245-t006_Table 6Table 6Characteristics and applications of biomaterials on bone tissue.MaterialsCharacteristicsApplicationsRef.AlginateFunctionalization with hydroxyapatite and chitosanImprovement of cell adhesion and proliferation.[167]ChitosanFunctionalization with adhesive peptidesImprovement of cell adhesion and proliferation.[165]PCLFunctionalization organic and superparamagnetic nanoparticlesVascularization of bone scaffold Improvement of cell adhesion and proliferation[177,179,180]PLAFunctionalization with organic and natural nanoparticlesVascularization of bone scaffold Periapical and alveolar bone regeneration Exosome-enriched bone scaffold[181,182,183]TitaniumTuneable porosity Functionalization with coatingBone implants[183,184,185,186,187,188]Type I collagenBioactivity Functionalization3D-printed bone scaffold[169,170,171,172]Wollastonite diopsideFunctionalizationScaffold functionalized with adhesive peptides[166]

### 4.6. Cartilage and Tendon Regeneration

Cartilage has limited intrinsic self-repair capacity, therefore the effective treatment of its defects represents a challenging problem. The “Aldo Moro” Bari University has proposed composite scaffolds made of gellan gum, inorganic clay derived from mesoporous silica, and antibacterial Manuka honey. Polymeric scaffolds generally have poor mechanical properties, on contrary this mesoporous silica-composite hydrogels exhibited significant changes in peak elastic and dynamic moduli over time. Moreover, the scaffold showed in vitro cytocompatibility and antibacterial preventive capability [189]. Instead, the Scuola Superiore Sant’Anna of Pisa has explored the properties of JellaGel™, composed by jellyfish material and genipin, a natural crosslinker. The collagen-based hydrogel is biocompatible and an attractive option for cartilage regeneration [190].

Blends of polymers have been studied to fabricate substitutes for tendon reconstruction. The ideal scaffold should have a structure mimicking the natural tendon while providing adequate mechanical strength and stiffness. Sensini et al. (Bologna University) developed electrospun nanofibers crosslinking PLLA and collagen. The nanofibers were then aligned and wrapped in bundles. Human fibroblasts seeded on the bundles had increased metabolic activity, meanwhile the stiffness, strength and toughness of the bundles were comparable to tendon fascicles [191]. In Teramo, electrospun PLGA fleeces with highly aligned fibers that mimic tendon ECM have been combined with amniotic epithelial stem cells (AECs). It was observed that fiber alignment influenced cell morphology determining the morphological change of AECs from a cuboidal to a fusiform tenocyte-like shape. Moreover, cell proliferation, tenogenic differentiation, immunomodulation, and fleece mechanical properties were regulated by changing the fiber diameter [192,193].

## 5. Conclusions

In conclusion, this review summarizes the recent efforts of Italian research centers in studying and developing regenerative biomaterials (Figure 7). In particular, we have provided a snapshot of the last five years of research with a particular focus on polymers. In future, polymers, both natural and synthetic, will find increasing use in the medical field. In fact, they can be the main component, or the coating and vehicle to functionalize ceramic or metallic biomaterials. Furthermore, the modification of polymer backbones allows for the tuning of the intrinsic mechanical properties and bioactivity. In light of these considerations, we presume that polymers can provide further important inputs for the development of new materials.

## Figures and Tables

**Figure 1 ijms-23-08245-f001:**
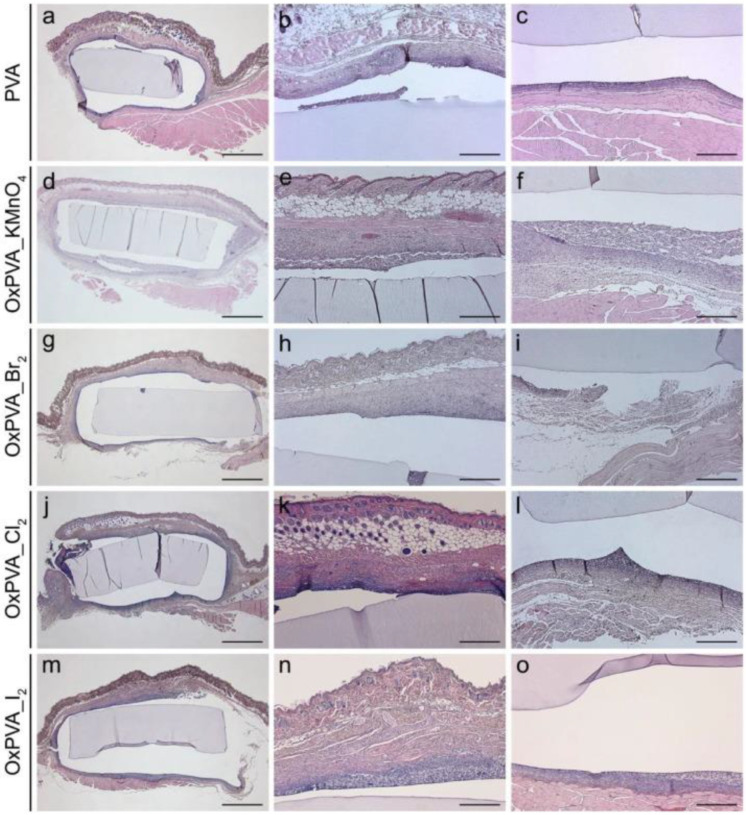
Histological evaluation of oxidized PVA scaffolds. Hematoxylin and eosin staining of neat and oxidized PVA scaffolds after four weeks of subcutaneous implantation into the dorsal region of BALB/c mice. (**a**,**d**,**g**,**j**,**m**) All hydrogel disks could be well identified among the surrounding tissues. (**b**,**e**,**h**,**k**,**n**) No severe inflammatory infiltration was observed at the subcutaneous and (**c**,**f**,**i**,**l**,**o**) muscular sides. Scale bars: (**a**,**d**,**g**,**j**,**m**) 100 µm; (**b**,**c**,**e**,**f**,**h**,**i**,**k**,**l**,**n**,**o**) 400 µm. OxPVA_KMnO_4_: PVA oxidized with potassium permanganate; OxPVA_Br_2_: PVA oxidized with bromine; OxPVA_Cl_2_: PVA oxidized with chlorine; OxPVA_I_2_: PVA oxidized with iodine. Reproduced with permission [17].

**Figure 2 ijms-23-08245-f002:**
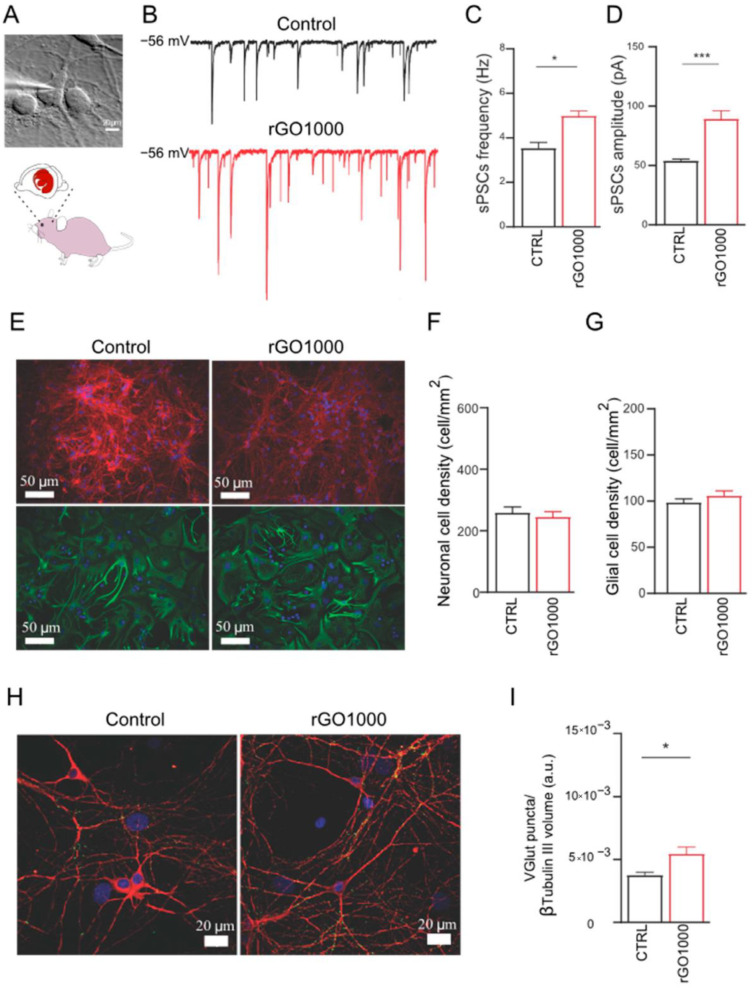
Impact of GO derivatives on the nervous system. (**A**) Neuronal cultures were obtained from rat brain after exposure to GO derivate (rGO1000) or vehicle (control). (**B**) Neuronal activity was monitored by patch-clamp (exemplificative voltage-clamp traces). (**C**) rGO1000 treated neurons presented a significant increase in spontaneous postsynaptic currents (sPSCs) frequency, and (**D**) amplitude. (**E**) Immunofluorescence images representative of neurons (β-tubulin III in red), glial cells (GFAP in green), and nuclei (DAPI in blue). (**F**) Neuronal and (**G**) glial densities in control and treated condition. (**H**) Confocal reconstructions of control and treated neurons stained for the vesicular glutamate transporter 1 (VGLUT1 in green) and for β-tubulin III (red); nuclei are stained with DAPI (in blue). (**I**) VGLUT1-positive puncta (synaptic activity) in rGO1000 treated cultures increased significantly. Reproduced with permission [54]. * *p* = 0.1; *** *p* = 0.005.

**Figure 3 ijms-23-08245-f003:**
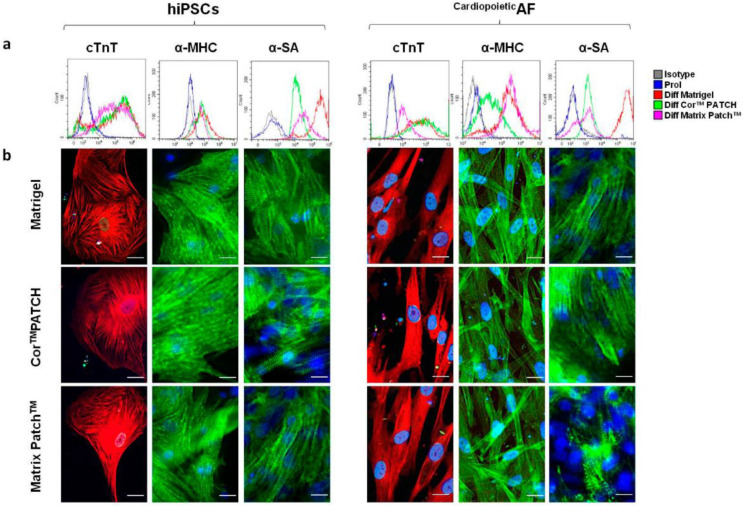
Decellularized extracellular matrices used in cardiac surgery can efficiently support the proliferation and cardiac differentiation of stem cells. Human induced pluripotent stem cells (hiPSCs) and human amniotic fluid cells that express the multipotency markers SSEA4, OCT4, and CD90 (^Cardiopoietic^AF) have been differentiated for 15 days on Cor^TM^ PATCH, Matrix Patch^TM^, or Matrigel^®^. The Cor^TM^ PATCH is a decellularized, porcine, intestinal submucosal extracellular matrix used for cardiovascular injury. The Matrix Patch^TM^ is a cell-free, equine-derived, pericardial patch. Matrigel^®^ is a substrate matrix used for in vitro tests but is not transplantable in humans. (**a**) A flow cytometry analysis of the mature cardiomyocytes hallmark proteins: cardiac troponin T (TnT), α-myosin heavy chain (α-MHC), and α-sarcomeric actin (α-SA). (**b**) Immunofluorescent staining for cTnT (red), α-MHC (green), and α-SA (green) in hIPSCs and ^Cardiopoietic^AF cells on Core^TM^ PATCH, Matrix Patch^TM^, or Matrigel^®^, as indicated. The nuclei were counterstained with DAPI (blue). Scale bar: 10 μm. Reproduced with permission [110].

**Figure 4 ijms-23-08245-f004:**
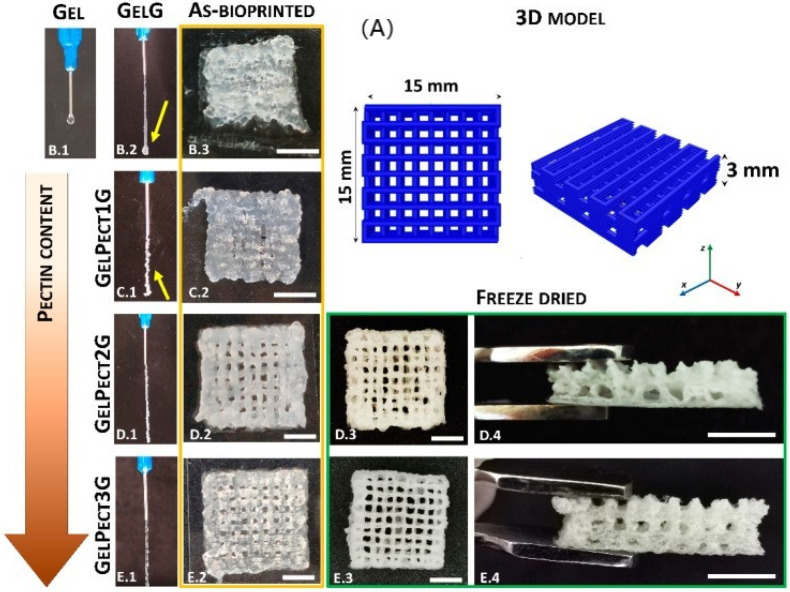
Patch biofabrication: from 3D model to production. (**A**) Illustration of the 3D model of a woodpile scaffold. Extrusion of various gelatin-based bio-ink: (**B.1**) gelatin (GEL) without crosslinking agent; (**B.2**) gelatin (GELG) with crosslinking agent GELG; (**C.1**) bio-ink gelatin-pectin (5–0.5%) (GELPect1G); (**D.1**) bio-ink gelatin-pectin (0.1–1%) (GELPect2G); (**E.1**) bio-ink gelatin-pectin (5–2.5%) (GELPect3G). (**B.3**,**C.2**,**D.2**,**E.2**) Pictures of 3D woodpile structures as bioprinted and (**D.3**,**D.4**,**E.3**,**E.4**) as freeze-dried (scale bars = 5 mm). Reproduced with permission [120].

**Figure 5 ijms-23-08245-f005:**
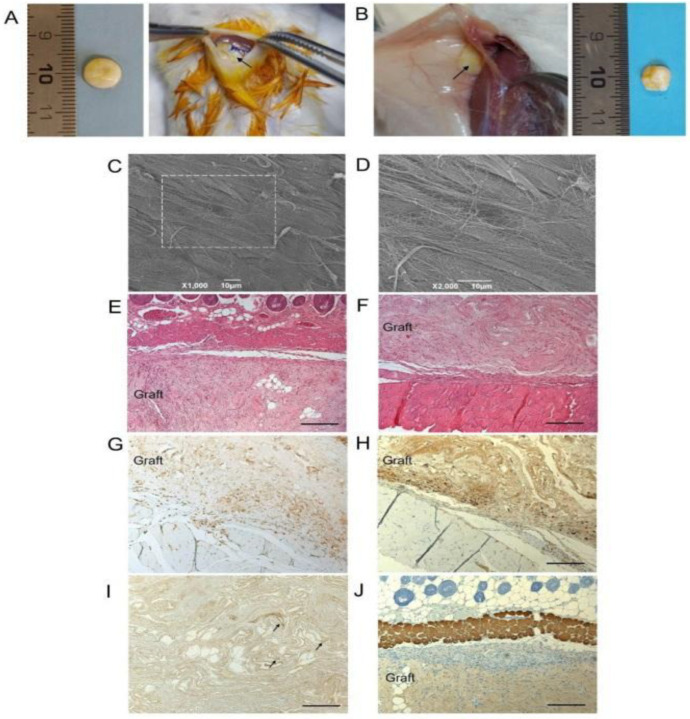
Decellularized human diaphragm integration in host tissue. (**A**) Discoidal sample was implanted subcutaneously anchored to the dorsal muscle of BALB/c mice. (**B**) Sample excision after 14 days. (**C**,**D**) Ultrastructural analysis by SEM. (**E**) Histological staining with haematoxylin and eosin at the graft–host interface on the subcutaneous and (**F**) deeper muscular side. The immunolocalization of (**G**) CD3+ and (**H**) F4/80+ cells at the boundaries between the graft and the host tissue confirmed moderate lympho-monocytic infiltration triggered by the scaffold graft. (**I**) Positivity to VEGF (black arrows) and (**J**) negativity to myosin within the diaphragmatic graft demonstrated an early angiogenetic, but not myogenetic, process after 14 days from implantation. Scale bar: 10 µm (**C**,**D**); 200 µm (**E**–**J**). Reproduced with permission [134].

**Figure 6 ijms-23-08245-f006:**
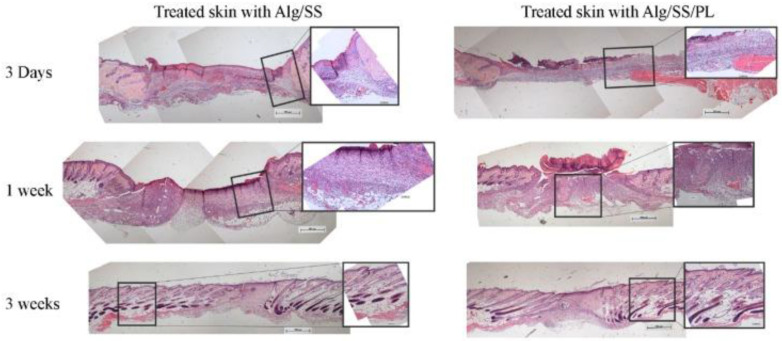
Full-thickness skin lesions in a mice model treated with alginate-based membranes. Haematoxylin-eosin staining of skin lesions treated with alginate-silk sericin sponge (Alg/SS) (**left**) and with alginate-silk sericin-platelet derivative sponge (Alg/SS/PL) (**right**). After 3 weeks, the lesion treated with Alg/SS/PL sponge showed a more organized tissue similar to a healthy skin. Reproduced with permission [141].

**Figure 7 ijms-23-08245-f007:**
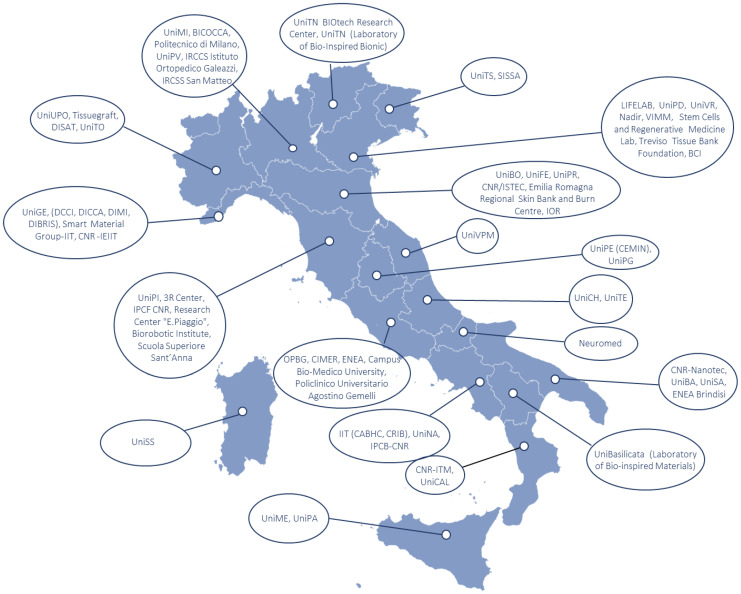
Distribution of the main research Italian centers for biomaterial production and testing.

**Table 4 ijms-23-08245-t004:** Comparison between conventional treatments and novel treatment.

Conventional Treatment	Novel Treatment
Skin Grafting with Autograft: autograft is derived from the patient’s own tissue	Cell co-cultures: keratinocytes and fibroblast are co-cultured for tissue generation
Skin Allograft: allografts obtained from donors	Cultured Epithelial Autografts: sheets of keratinocyte cells are cultured on mouse fibroblasts
Xenograft: surgical graft from one species to another dissimilar species	Tissue-engineered skin substitutes: skin substitute preparation involves the cells and/or the extracellular matrix. Different approaches have been adopted to develop engineered tissues, such as synthetic membranes for mono- or multi-layered cultures and 3D matrices for full-thickness models
Amnion: Amnion collected from the placentae of selected and screened donors. The amniotic membrane is rich in collagen and several growth factors that support the healing process to both advance wound closure and diminish scar formation	

**Table 5 ijms-23-08245-t005:** Characteristics and applications of biomaterials on skin tissue.

Materials	Characteristics	Applications	Ref.
acellular dermal matrice	Decellularized matrix	Biological scaffolds	[143,144]
Alginate/silk sericin/platelet lysate	Bioactive freeze-dried sponge	Formation of granulation tissue and deposition of new collagen	[141]
platelet lysate/PRP	Bioactive molecules derived from the activation of a platelet concentrate in the presence of cryoprecipitate, thrombin and calcium gluconate	Functionalize membranes	[141,143]
PLGA	Low toxicity	Devices in contact with the skin	[145]
PLLA/PLGA	Biodegradable	Drug delivery system	[146,147,148]
PEGDA	Biocompatible	Sensors and drug delivery systems	[154,155,156,157,158,159,160,161,162]

## Data Availability

Data is contained within the article.

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
