# Peer review of "Biomaterials for Regenerative Medicine in Italy: Brief State of the Art of the Principal Research Centers"

_ijms, 2022, doi:10.3390/ijms23158245_

Round 1
Reviewer 1 Report
I've read with interest the review article from Camponogara et al, entitled "Biomaterials for regenerative medicine in Italy: brief state of the art of the principal research centers".
They authors provide a large and quite complete scope of current regenerative medicine research strategies in Italy. It appears like a potential very useful information, but i feel like it is more relevant to be provided as public research report (Italian National Research Agency or EU research), rather than a review scientific paper in an international scientific journal: indeed, it is an exhaustive and very large list and juxtaposition of italian RM research centers activities and publications, rather than a targeted and articulated review.
Just an important remark in the introduction: there is a confusion between regenerative medecine and tissue engineering - indeed tissue engineering is a subgroup of regenerative medicine. I.e. cell therapy can be used as an organ regenerative tool, but doesn't employ tissue engineering solutions. Second concept remark: Regenerative Medicine is not only targeting "wound healing", as stated in the introduction.
Author Response
Just an important remark in the introduction: there is a confusion between regenerative medecine and tissue engineering - indeed tissue engineering is a subgroup of regenerative medicine. I.e. cell therapy can be used as an organ regenerative tool, but doesn't employ tissue engineering solutions. Second concept remark: Regenerative Medicine is not only targeting "wound healing", as stated in the introduction.
thanks, we modified according
Reviewer 2 Report
The manuscript titled “Biomaterials for regenerative medicine in Italy: brief state of the art of the principal research centers” is a review of the literature presenting the main research advances on natural and synthetic polymers which are under investigation in Italy for tissue engineering (TE) purposes.
It is an interesting overview on polymeric biomaterial research carried out by the main Italian Centers with the aim of finding novel regenerative strategies for a variety of tissues (i.e., neural and cardiovascular tissues, skin, bone). The manuscript is well written and easy to read, with appropriate English language and grammar.
Given the quality of the work, below are some concerns that still need to be addressed:
- the Abstract should be a little more extensive, giving more detailed background about the topic of the review article and better specifying the reason why this paper should be scientifically useful, which seems now only implicit. The aim of the work should be clearly declared also in the Introduction section.
- a part of the literature seems to be missing about the biomaterial research in Padua University. In particular, the following synthetic, natural and composite materials have been largely researched in Padua for the following TE applications:
1) polyvinyl alcohol (PVA) hydrogel was assessed as (a) a biomimetic scaffold with enhanced properties for tissue regeneration (i.e., Stocco et al., 2017, doi: 10.1002/term.2101; Barbon et al., 2020, doi: 10.3390/ijms21030801); (b) an ideal support/drug-delivery material for nerve conduits/wraps fabrication (Stocco et al., 2018, doi: 10.1038/s41598-017-19058-3; 2019, doi: 10.1038/s41598-019-53812-z; 2021- doi: 10.3390/polym13193372; Porzionato et al., 2019, doi: 10.3390/ma12121996; arbon et al., 2016, doi: 10.1016/j.taap.2016.09.001) and vessel (Conconi et al., 2014, doi: 10.3892/mmr.2014.2348) replacement; (c) the synthetic part of biohybrid scaffolds for cartilage (Stocco et al., 2014, doi: 10.1155/2014/762189; 2016, doi: 10.1007/s00441-016-2408-8) and gut (Grandi et al., 2018, doi: 10.1155/2018/7824757) regeneration;
2) Poly-ε-caprolactone composite scaffolds were developed for bone repair in vitro studies (Di Liddo et al., 2014, doi: 10.3892/ijmm.2014.1954);
3) decellularized matrices was developed and tested both in vitro and in vivo for skeletal muscle (i.e., Conconi et al., 2005, doi: 10.1016/j.biomaterials.2004.07.035; De Coppi et al., 2006, doi: 10.1089/ten.2006.12.1929; Porzionato et al., 2015, doi: 10.3390/ijms160714808; Boso et al., 2021, doi: 10.3390/biomedicines9070709; Barbon et al., 2022, doi: 10.3390/biomedicines10040739), vessel (Porzionato et al., 2017, doi: 10.1166/jbt.2017.1545: Grandi et al., 2011, doi: 10.3892/ijmm.2011.720), trachea (Conconi et al., 2005, doi: 10.1111/j.1432-2277.2005.00082.x), esophagus (i.e., Marzaro et al., 2006, doi: 10.1002/jbm.a.30666) and heart valve (Di Liddo et al., 2016; Iop et al., 2017, doi: 10.1007/s12265-017-9738-0) regeneration
4) platelet-rich fibrin was considered as a biological material which could serve as a cell and drug-delivery platform for regenerative purposes (i.e., Di Liddo et al., 2018; Barbon et al., 2018, doi: 10.1002/term.2713)
Some of the studies mentioned above have also been published within the LifeLab Research Program funded by the Veneto Region, which was mentioned in the manuscript. These studies should also be reported among the research supported by the Program.
- Related to the previous point, it appears that the literature review should be more carefully performed, according to systematic methods/criteria which should be also described in detail by adding a Methodology section within the text.
- a subparagraph focused on Cartilage application should be considered separated from the Bone paragraph.
- Some figures are missing to complete the text. Experimental figures may be taken from published works included in the Review, respecting copyright issues.
Author Response
thanks to the referee.
- the Abstract should be a little more extensive, giving more detailed background about the topic of the review article and better specifying the reason why this paper should be scientifically useful, which seems now only implicit. The aim of the work should be clearly declared also in the Introduction section.
- thanks done
- a part of the literature seems to be missing about the biomaterial research in Padua University. In particular, the following synthetic, natural and composite materials have been largely researched in Padua for the following TE applications:
- thanks done
1) polyvinyl alcohol (PVA) hydrogel was assessed as (a) a biomimetic scaffold with enhanced properties for tissue regeneration (i.e., Stocco et al., 2017, doi: 10.1002/term.2101; Barbon et al., 2020, doi: 10.3390/ijms21030801); (b) an ideal support/drug-delivery material for nerve conduits/wraps fabrication (Stocco et al., 2018, doi: 10.1038/s41598-017-19058-3; 2019, doi: 10.1038/s41598-019-53812-z; 2021- doi: 10.3390/polym13193372; Porzionato et al., 2019, doi: 10.3390/ma12121996; arbon et al., 2016, doi: 10.1016/j.taap.2016.09.001) and vessel (Conconi et al., 2014, doi: 10.3892/mmr.2014.2348) replacement; (c) the synthetic part of biohybrid scaffolds for cartilage (Stocco et al., 2014, doi: 10.1155/2014/762189; 2016, doi: 10.1007/s00441-016-2408-8) and gut (Grandi et al., 2018, doi: 10.1155/2018/7824757) regeneration;
2) Poly-ε-caprolactone composite scaffolds were developed for bone repair in vitro studies (Di Liddo et al., 2014, doi: 10.3892/ijmm.2014.1954);
3) decellularized matrices was developed and tested both in vitro and in vivo for skeletal muscle (i.e., Conconi et al., 2005, doi: 10.1016/j.biomaterials.2004.07.035; De Coppi et al., 2006, doi: 10.1089/ten.2006.12.1929; Porzionato et al., 2015, doi: 10.3390/ijms160714808; Boso et al., 2021, doi: 10.3390/biomedicines9070709; Barbon et al., 2022, doi: 10.3390/biomedicines10040739), vessel (Porzionato et al., 2017, doi: 10.1166/jbt.2017.1545: Grandi et al., 2011, doi: 10.3892/ijmm.2011.720), trachea (Conconi et al., 2005, doi: 10.1111/j.1432-2277.2005.00082.x), esophagus (i.e., Marzaro et al., 2006, doi: 10.1002/jbm.a.30666) and heart valve (Di Liddo et al., 2016; Iop et al., 2017, doi: 10.1007/s12265-017-9738-0) regeneration
4) platelet-rich fibrin was considered as a biological material which could serve as a cell and drug-delivery platform for regenerative purposes (i.e., Di Liddo et al., 2018; Barbon et al., 2018, doi: 10.1002/term.2713)
Some of the studies mentioned above have also been published within the LifeLab Research Program funded by the Veneto Region, which was mentioned in the manuscript. These studies should also be reported among the research supported by the Program.
- Related to the previous point, it appears that the literature review should be more carefully performed, according to systematic methods/criteria which should be also described in detail by adding a Methodology section within the text.
- thanks done
- a subparagraph focused on Cartilage application should be considered separated from the Bone paragraph.
- thanks done
- Some figures are missing to complete the text. Experimental figures may be taken from published works included in the Review, respecting copyright issues.
- thanks added
Reviewer 3 Report
Dear authors.
The presented publication is well prepared and may help to develop international cooperation in the field of regenerative medicine. Thanks to this publication, I have noticed the potential of Italian universities to work on tissue and cell research. Until now, I have only associated the preparation of Holoclar. When I was in Italy, for example in Pisa, I would pass university buildings and ask myself about the research being done there. And suddenly the answer appeared.
I accepted the work without comment.
Author Response
many thanks to the referee